# Gene expression QTL mapping in stimulated iPSC-derived macrophages provides insights into common complex diseases

Nikolaos I. Panousis[1], Omar El Garwany[1], Andrew Knights[1], Jesse Cheruiyot Rop [1], Natsuhiko Kumasaka[1], Maria Imaz [1,2], Lorena Boquete Vilarino[1,3], Anthi Tsingene[1], Alex Tokolyi[1], Alice Barnett [1], Celine Gomez[1], Carl A. Anderson [1,4] ✉ & Daniel J. Gaffney [1,4]

Many disease-associated variants are thought to be regulatory but are not present in existing catalogues of expression quantitative trait loci (eQTL). We hypothesise that these variants may regulate expression in specific biological contexts, such as stimulated immune cells. Here, we used human iPSC-derived macrophages to map eQTLs across 24 cellular conditions. We found that 76% of eQTLs detected in at least one stimulated condition were also found in naive cells. The percentage of response eQTLs (reQTLs) varied widely across conditions (3.7% – 28.4%), with reQTLs specific to a single condition being rare (1.11%). Despite their relative rarity, reQTLs were overrepresented among disease-colocalizing eQTLs. We nominated an additional 21.7% of disease effector genes at GWAS loci via colocalization of reQTLs, with 38.6% of these not found in the Genotype–Tissue Expression (GTEx) catalogue. Our study highlights the diversity of genetic effects on expression and demonstrates how condition-specific regulatory variation can enhance our understanding of common disease risk alleles.

Disease-associated variants are often located in noncoding regions of the genome[1,2], making biological interpretation of their function challenging. If these variants alter gene expression this should, in principle, be detectable by mapping expression quantitative trait loci (eQTL). eQTLs have now been discovered across a broad range of human tissues and cell types[3,4], with at least one eQTL known for every human gene. Despite this, a surprisingly large fraction of disease associations do not share a causal variant with a known eQTL[5–8]. For example, the most recent publication by the Genotype-Tissue Expression (GTEx) consortium, which mapped eQTLs across 49 tissues ascertained from 838 individuals, reported that approximately 43% of disease associations colocalize with a detectable eQTL[3].

One potential explanation for missing disease-associated eQTLs is that many regulatory variants may function in highly specific cellular contexts, such as activated immune cells. To address this, multiple studies have now been performed using bulk RNA-sequencing of activated cell conditions[9–17] and more recently using single cell approaches[18,19]. However, because primary cell material is frequently limited, eQTL mapping has typically been restricted to a relatively small set of environmental contexts following perturbation with a limited set of classical stimuli.

Here, we differentiated induced pluripotent stem cells (iPSCs) from 209 individuals to macrophages and mapped eQTLs across twelve different cellular conditions at two timepoints post stimulation. We used this dataset (MacroMap, https://www.macromapqtl.org.uk/)

[1]Wellcome Sanger Institute, Wellcome Genome Campus, Hinxton, UK. [2]Department of Public Health and Primary Care, University of Cambridge, Cambridge, UK. [3]Centro de Fabricación de Terapias Avanzadas, A Coruña, Spain. [4]These authors jointly supervised this work: Carl A. Anderson, Daniel J. Gaffney. ✉e-mail: ca3@sanger.ac.uk

to explore the properties of response eQTLs (reQTLs) and their relevance for disease.

## Results

### Expression profiling of macrophage innate immune responses

We selected 217 iPSC lines derived from unrelated healthy donors by the HipSci consortium[20], and differentiated them to macrophages using a previously described protocol[15] with minor modifications (see Methods and Supplementary note).

During the differentiation process, we collected macrophage precursor cells at day 0 (labelled "Prec_D0") and day 2 ("Prec_D2"). We used a low-input RNA-seq protocol to profile the transcriptomes of these cells, as well as those of unstimulated iPSC-derived macrophages after 6 and 24 h. We next perturbed the cells with a panel of ten different stimuli and measured gene expression six and 24 h after stimulation using the same protocol (Fig. 1a). Our stimulation panel (Supplementary Data 1) included stimuli that trigger pro- and anti-

inflammatory pathways in macrophages (IFNβ, IFNγ, interleukin-4/IL4) and those that induce response to viral infection (Resiquimod/R848, Poly I:C/PIC). We also included combinations of stimuli to mimic bacterial response to infection (smooth lipopolysaccharide/sLPS, Pam3CSK4/P3C, CD40 ligand + IFNγ + sLPS/CIL, interleukin-10 + sLPS/ LIL10). Furthermore, we stimulated macrophages with myelin basic protein (MBP) to mimic the innate immune response of microglia (brain-resident macrophages) to stimuli. For simplicity throughout, we include day 0 and day 2 precursor cells when we refer to "stimulated conditions", except where otherwise stated. We profiled the transcriptome of 5208 samples and after quality control ("Methods") retained 4698 unique RNA-seq libraries from 209 unique iPSC lines (Supplementary Fig. 1).

We first explored which technical and biological confounders affected gene expression using variance component analysis (Supplementary Fig. 1d) and identified that the combined factors of stimulation type and differentiation time, as the second most important

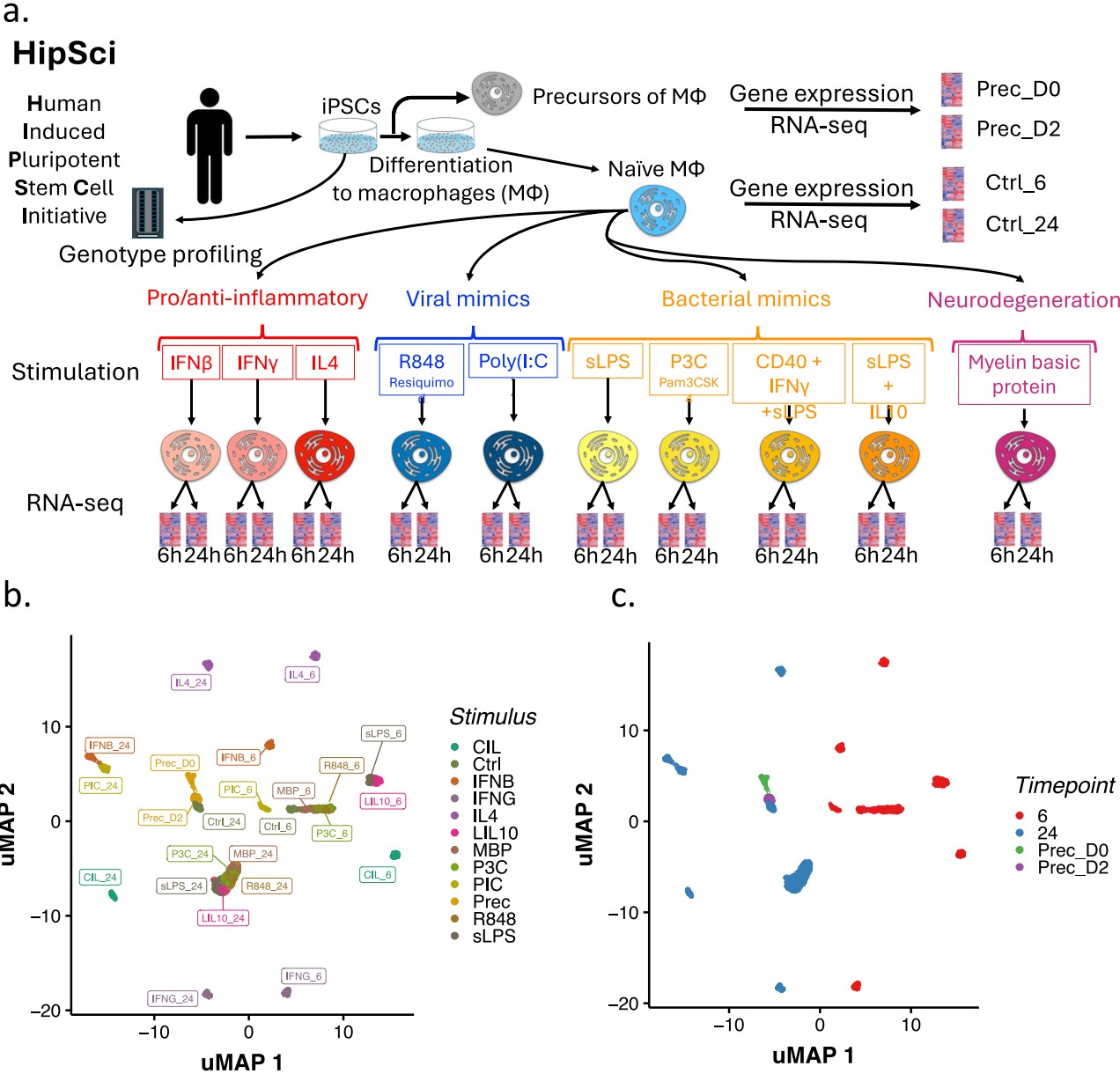

**Fig. 1 | Experimental workflow and transcriptomic analysis. a** Overview of the experimental workflow. Isolation and stimulation of naïve macrophages and measurement of gene expression via RNA-seq at two post-stimulation time points (6 h and 24 h) for multiple stimuli. UMAP representation of 4698 RNA-seq libraries after quality control ("Methods") coloured by different stimulation conditions (**b**) and time point (**c**).

driver of the total expression variation (29.3%) after the library preparation method (39.9%). Notably, the library preparation method variation is primarily due to the choice of medium used during myeloid precursor formation (Supplementary Fig. 1e).

A UMAP projection of the gene expression data showed that samples clustered by stimulation and time point (Fig. 1b, c). To better quantify the structure present in the gene expression data we estimated the pairwise Pearson's correlation ($r$) between conditions (mean TPM values per gene across all individuals for a given condition) (Supplementary Fig. 2a). For example, the gene expression profile of precursor cells at day 2 clustered closely with samples from naive cells at the 24 h time point (Pearson $r = 0.98$). Likewise, gene expression in macrophages stimulated with PIC was highly correlated with gene expression in IFNβ at both time points, highlighting an expected overlap in signalling because PIC increases IFNβ expression[21].

We next identified differentially expressed genes (DEGs) between naive and stimulated conditions using DESeq2[22] ("Methods"). On average, we found a median of 2306 DEGs (false discovery rate (FDR) = 5%, fold change ≥2) between naive and stimulated cells across all conditions (Supplementary Data 2–4), although the number varied widely between conditions (488–5427 DEGs) (Supplementary Fig. 2b). These DEGs were enriched in gene ontology (GO) terms (Supplementary Data 5 and 6) and Reactome pathways[23] (Supplementary Data 7 and 8) corresponding to relevant immunological pathways. For example, "response to bacterium" (GO:0009617) was the most enriched GO term in DEGs 6 h after sLPS stimulation, with *TNF* upregulated 13.75-fold[24]. Likewise, "Response to virus" (GO:0009615) was the GO term most enriched with genes differentially expressed in cells 6 h after stimulation with IFNβ, with *GBP1* (Guanylate Binding Protein 1) upregulated 256-fold[25–27]. This demonstrates that iPSC derived macrophages can faithfully recapitulate the known biological pathways activated by different stimuli.

## Genetic regulatory effects across different stimulation conditions

To investigate genetic regulation of gene expression we mapped expression quantitative loci (eQTL) across all twenty four conditions (Methods). We mapped cis-eQTLs within ±1Mbp of the transcription start site (TSS) of each gene at a false discovery rate (FDR) of 5%. We found between 1781–3735 eGenes (genes with at least one cis-SNP significantly associated with expression) per condition (Supplementary Fig. 3a), with an enrichment of eQTL lead SNPs (eSNPs) in close proximity to promoters ($p = 1.2 \times 10^{-124}$, mean log odds ratio = 2.29) (Supplementary Fig. 3b). We performed conditional eQTL mapping to identify additional cis-eQTLs with independent effects on expression ("Methods") and found that 3.3%–8% of eGenes in each of our 24 cell conditions, and 18% of all eGenes, had more than one eSNP (1817 eGenes) (Supplementary Fig. 4a). Secondary and tertiary eSNPs (conditionally independent eQTLs) were further away from the TSS (mean distance 206 and 233 kb respectively) compared to the primary signals (mean distance 94 kb Wilcoxon $p = 1.1 \times 10^{-215}$, $p = 5.7 \times 10^{-16}$ respectively) (Supplementary Fig. 4b).

Per condition, our eQTL detection rate (22.4% of tested genes, mean over all conditions) was lower compared to GTEx tissues of similar sample size (35% in GTEx tissues of between 180–210 individuals). One likely explanation for our lower detection rate is that we studied a single cell type, while most GTEx eQTL studies are derived from complex tissues. Consistent with this, GTEx eQTLs in lymphoblastoid cell lines (LCLs), had an equivalent detection rate to our study (~22%). Summing over all cell conditions we identified 10,170 unique eGenes (72.4% of expressed genes) (Fig. 2a). This is higher than expected given our study sample size (mean across all conditions $n = 202$), even compared to studies of complex tissues. For example, the number of eGenes discovered in GTEx liver ($n = 208$) and brain cortex ($n = 205$) was 4415 and 7108, equivalent to 25.6% and 37.5% of

expressed genes, respectively. Our overall eQTL detection rate also exceeds that observed for individual cell types studied in GTEx. For example, we detected a similar proportion of eQTLs to that found in GTEx cultured fibroblasts (12,280 eGenes, 73.2% of expressed genes), a study with over double our sample size ($n = 483$). Although studying many conditions likely revealed additional eQTLs, the large number of eQTLs we detected is likely also driven by the very high degree of biological replication in our study. We note that even when we compared our control conditions at 6 and 24 h timepoints, we detected an additional 774 new eGenes (Supplementary Fig. 4c). Thus, generating additional gene expression data from the same individuals increased our power to detect eQTLs relative to other studies of similar sample size.

## Stimulation specific genetic regulation of gene expression in immune response

To better establish which cis-eQTLs were truly restricted to stimulated cells, we used mashr[28], which compares eQTL effect size estimates between conditions. Using the "common baseline" mode ("Methods"), mashr estimates the extent to which the eQTL effect size in each stimulated condition deviates from a baseline condition, here defined as the Ctrl_24 condition. Mashr produces a local false sign rate (lfsr) that measures the confidence in the direction of each genetic effect compared to the baseline effect[29].

We defined a response eQTL (reQTL) as a significant difference (lfsr < 0.05) in genetic effect between the baseline condition and at least one stimulated condition. The number of reQTLs we detected varied widely between conditions, from 3.7% of all eQTLs in precursor cells at day 2 (Prec_D2) to 28.4% of all eQTLs in cells stimulated with CIL at the 6 h time point (Fig. 2b). Across all conditions, 23.4% (2378) of eGenes had a reQTL in at least one condition with the majority of these (21.9%, 2228) having a larger absolute effect following stimulation than in the naive condition. Approximately 9% (159) of conditionally independent eQTLs were classified as reQTLs. 89% (142) of these genes with a conditionally independent reQTL also had a primary reQTL, an almost 4-fold enrichment (Fisher's exact test $p$ value = $3^{-36}$, OR = 12.93, 95% CI 7.7–23.05) (Supplementary Fig. 5b).

Our ability to detect eQTLs for lowly expressed genes is limited. Many of our reQTLs may therefore have similar effect sizes in both the naive and stimulated conditions, but these effects can only be detected as different once gene expression increases in response to stimulation. We found that between 5.8%–48.4% of reQTL genes were differentially expressed (example signals Fig. 2c, d) (FDR 5%, fold change ≥2) (31.2% on average across conditions), the majority of which (mean across conditions 75.8%) were up-regulated (Fig. 2c) compared to the naive condition. Nonetheless, the majority of reQTLs we identified (51.6%–94.2%) were in genes where we were unable to detect a substantial change in expression between the naive and stimulated conditions based on our chosen criteria (Fig. 2e).

Next, we asked how widely shared reQTLs were between stimulated conditions, using mashr. Very few (1.1%) reQTLs were specific to a single condition (condition-specific reQTLs), 90.5% of reQTLs were shared in five or more conditions, with 49% detected in at least half of the conditions (Fig. 3a, b). This suggests that the majority of the reQTLs we identified could have been detected with a smaller set of stimulated states.

To investigate the stability of reQTLs over time, we conducted pairwise comparisons between the 6-h and 24-h time points within each cell condition, as well as between day 0 and day 2 for precursor cells. To define time point specificity, we examined whether the reQTLs identified in the initial condition (6 h, lfsr ≤0.05) were absent in the second condition (24 h, lfsr >0.05) and vice versa. We observed substantial variation in time-specificity across conditions. For example, cells stimulated with PIC or IFNβ had relatively few time point-specific reQTLs (18–23.4%), perhaps reflecting permanent changes in

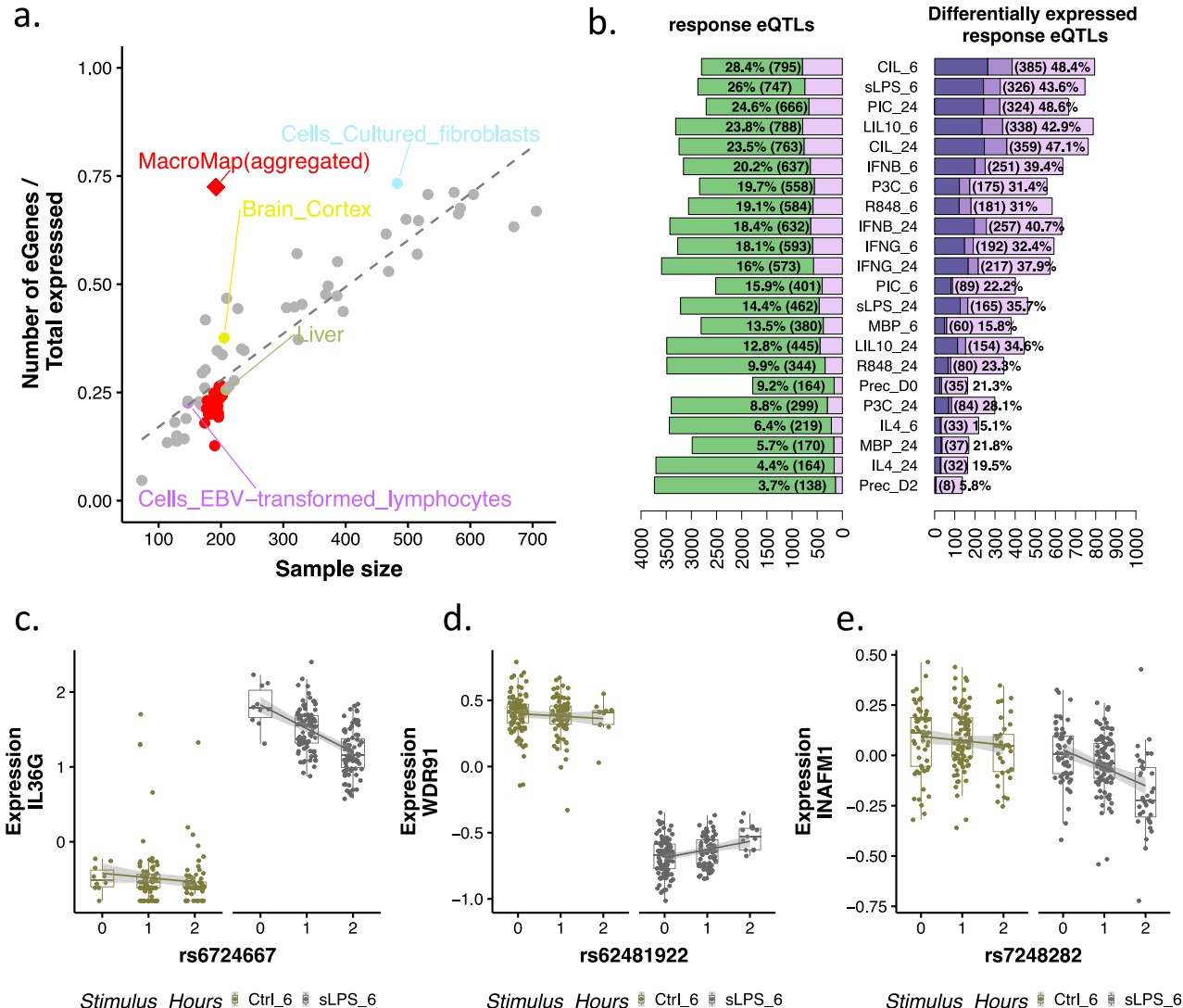

**Fig. 2 | Stimulation-induced changes in gene expression: identification and characterisation of reGenes in multiple conditions. a** Fraction of expressed genes (protein coding and lincRNAs) that are eGenes (genes with at least one eQTL) in our study (red diamond) and in different GTEx tissues (circles) as a function of the sample size (mean sample size across all conditions in our study). GTEx studies of single cell types (cultured fibroblast and lymphocytes) and tissues with similar sample size to our study (Brain-cortex and Liver) are highlighted. **b** Number and fraction of reGenes (pink) and the total number of eGenes (green) per stimulation condition (left hand panel) and number and fraction of reGenes that were differentially expressed (dark purple, up-regulated, dark pink, down-regulated) and not differentially expressed (pink) (right hand panel). Examples of genes where reQTL was detected following stimulation and genes were significantly up- (**c**), down-regulated (**d**) or where there was no significant change in expression (**e**) in the stimulated condition (sLPS at 6 h, n = 191 vs. Ctrl at 6 h, n = 179). Data are presented as individual points, jittered for visibility. Box plots represent the median (centre line), the first and third quartiles (bounds of the box), and the whiskers, which extend to the smallest and largest values within 1.5x the interquartile range from the bounds of the box. Data points outside this range are plotted as outliers. Shaded areas represent linear regression lines fitted to the data, with 95% confidence intervals.

chromatin state produced by IFNβ stimulation[30]. In contrast 85% of reQTLs detected in the precursor cells were specific to day, likely reflecting the very substantial changes in transcriptional regulation that occur during cell differentiation (Supplementary Fig. 5c).

The GTEx consortium has completed the most extensive and widely used study of eQTLs to date[3]. The vast majority of GTEx samples were post-mortem tissues and it is unclear how suitable these are for detecting condition-specific genetic effects in immune response. To investigate this, we tested whether the reQTLs detected in our study were also found in GTEx, estimating the overlap using the π1 statistic[31] (Supplementary Fig. 6). On average, half (54%) of the reQTLs detected in a given condition are replicated in a GTEx tissue, significantly lower than the replication rate of non-response eQTLs (63%) (Supplementary Fig. 6c, d) (Wilcoxon $p = 4.9 \times 10^{-47}$), with the highest replication rates observed for GTEx whole blood (Supplementary Fig. 6b). This suggests

that many of the immune pathways we have activated in our study are, at some level, also active in post-mortem tissue.

**Stimulation specific genetic regulation in disease**

We next investigated the relevance of reQTLs in disease. We collated a set of 83 well powered (ten or more genome-wide significant loci ($p \leq 5.0 \times 10^{-8}$) GWAS including 22 immune-mediated, 13 blood-related, 3 cancer, 11 cardiovascular, 15 neurological and 19 other traits or diseases (Supplementary Data 9). We found evidence of colocalization[32] between a disease association signal and an eQTL for 1955 (Supplementary Data 10) unique eGenes across all traits and conditions (posterior probability of sharing a single causal variant PP4 > 0.75). Relative to the naive conditions, stimulation often increased the strength of evidence of colocalization with disease. We found that including stimulated states substantially boosted our

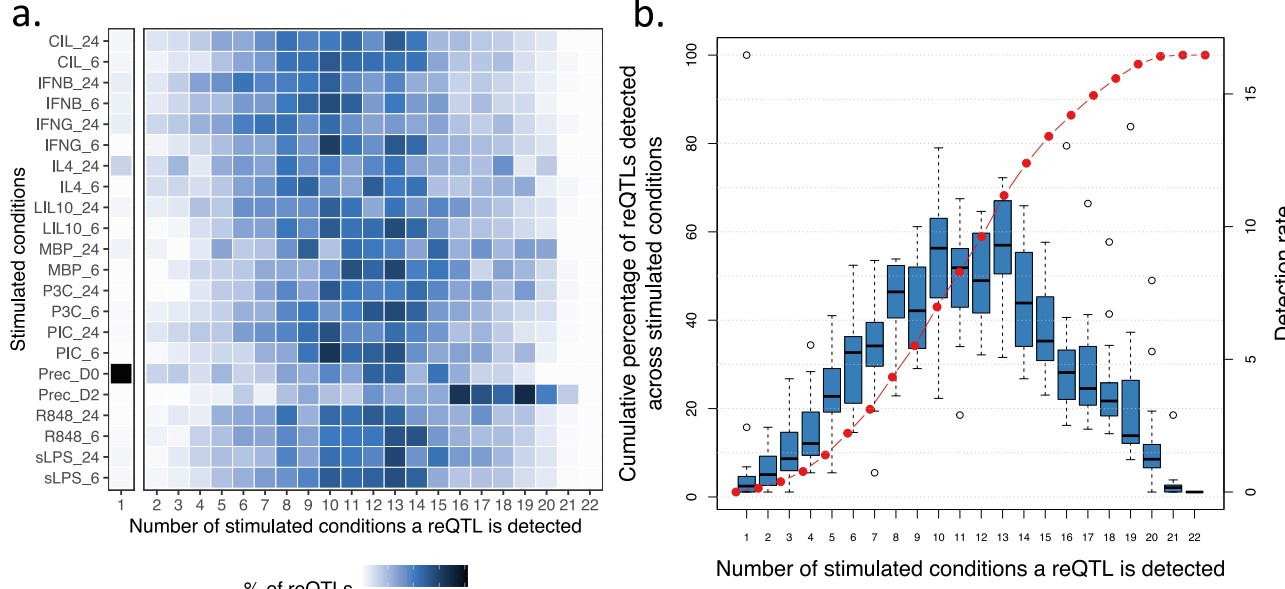

**Fig. 3 | Condition-specific and shared patterns of reQTLs across multiple stimuli. a** Proportions of reQTLs for every stimulated condition that were found in a single condition only (column 1) or detected in 2 or more stimulated conditions (columns 2–22). Colour intensity reflects the percentage of reQTLs in a given condition that were also detected in at least one other stimulated condition. Columns represent the number of conditions in which an reQTL was detected. **b** Cumulative percentage of detected reQTLs (red line and points, left hand *y* axis) and reQTL detection rate (blue boxplots, right hand *y* axis) versus the number of conditions in which an reQTL was detected. Box plots represent the median (centre line), the first and third quartiles (bounds of the box), and the whiskers, which extend to the smallest and largest values within 1.5x the interquartile range from the bounds of the box. Data points outside this range are plotted as outliers. The cumulative percentage of detected reQTLs (red line, left-hand *y*-axis) was calculated as the running sum of the mean detection rate across conditions. The *x*-axis indicates the number of stimulated conditions in which a reQTL was detected, and the right-hand *y*-axis shows the detection rate. All box plots are based on *n* = 22 stimulated conditions.

discovery of disease-eQTL colocalisations with only 32% of all colocalisations (631 eGenes) detected in naive cells (Fig. 4a).

The rise in the number of eGenes with colocalization evidence can be attributed to both the inclusion of more conditions relevant to disease and the increased power due to the substantial level of biological replication in our study. To investigate this further, we examined the frequency of disease-colocalized eQTLs that had significantly different effect sizes in naive and stimulated conditions using mashr. We found that 21.7% (424/1955; Fig. 4b, c) of eGenes with colocalization evidence had at least one reQTL, and among these, 89.4%(379) had a larger effect size in the stimulated condition than in the corresponding naive condition. Disease-colocalizing eGenes showed a significant overrepresentation of reQTL genes (*p* = 0.05, Fisher's exact test). This suggests that a substantial number of disease-related eQTLs will only be detected in stimulated conditions, because their effect sizes in naive cells are too small.

Our analysis revealed specific traits, as depicted in (Fig. 4c), where the use of stimulated macrophages appeared to be more relevant compared to naive conditions. Furthermore, it is important to note that, despite no individual trait exhibiting a substantial increase in the number of eGenes with evidence of colocalization due to reQTLs, this underscores the significance of reQTLs in unravelling disease loci that cannot be explained by naive eQTLs even though their transformative impact on significantly reshaping our comprehension remains limited.

Next, to determine the effectiveness of MacroMap in identifying effector genes within GWAS loci, we compared our results to those obtained using GTEx data. We asked how many of the 1955 disease colocalizations we found in our study could not have been identified using GTEx eQTLs. We found that 998 eGenes (51% 988/1955) were colocalized with higher confidence in our dataset (PP4 > 0.75 in MacroMap versus PP4 < 0.5 across all GTEx tissues) (Supplementary Fig. 7), 164 (16.6% 164/988) of which were defined by mashr as reQTLs. All 164

genes were expressed (minimum TPM = 0.8, mean TPM across all tissues = 165) in at least one GTEx tissue, and had a detectable eQTL (qvalue ≤0.05) in one or more GTEx tissues.

### A response eQTL implicates increased expression of *CTSA* with increased risk of coronary artery disease

One such colocalization event was identified between the CAD-associated risk locus at 20q13.12[33,34] and a reQTL observed 24 h after stimulation with either sLPS or sLPS + IL10 (LIL10) (PP4 = 0.98 and PP4 = 0.99 respectively, Supplementary Fig. 8a, d) and not with Ctrl conditions (Supplementary Fig. 8b). The risk-increasing allele (rs3827066, C > T) was associated with increased expression of *CTSA* without the gene being differentially expressed after stimulation with either sLPS or LIL10 (Supplementary Fig. 8c). Moreover, this risk allele has also been associated with increased risk of Abdominal Aortic Aneurysm (AAA)[35], a common complication of vessel wall impairment caused by various predisposing factors including atherosclerosis. Despite these disease associations being known for several years, the disease effector gene in the region had remained elusive. For example, the Open Targets Genetics Portal (https://genetics.opentargets.org/) showed that there are 34 genes in the locus, none of which had a variant-to-gene score >0.3, with *CTSA* ranked as the seventh most likely effector gene in the region. The TSS of *CTSA* is located 67.24 kb upstream of rs3827066, with six other genes in closer proximity to the SNP. Data from a promoter capture Hi-C study of circulating immune cells demonstrated that rs3827066 lies in a distal regulatory element of *CTSA* or *NEURL2* in many immune cell types, including monocytes and macrophages[36]. Our reQTL and colocalization results suggest the disease associated variant acts via dysregulation of *CTSA* expression.

*CTSA* is a lysosomal protective protein with both intracellular and extracellular functions[37]. In the lysosome, it is essential for both the activity and protection of beta-galactosidase and neuraminidase

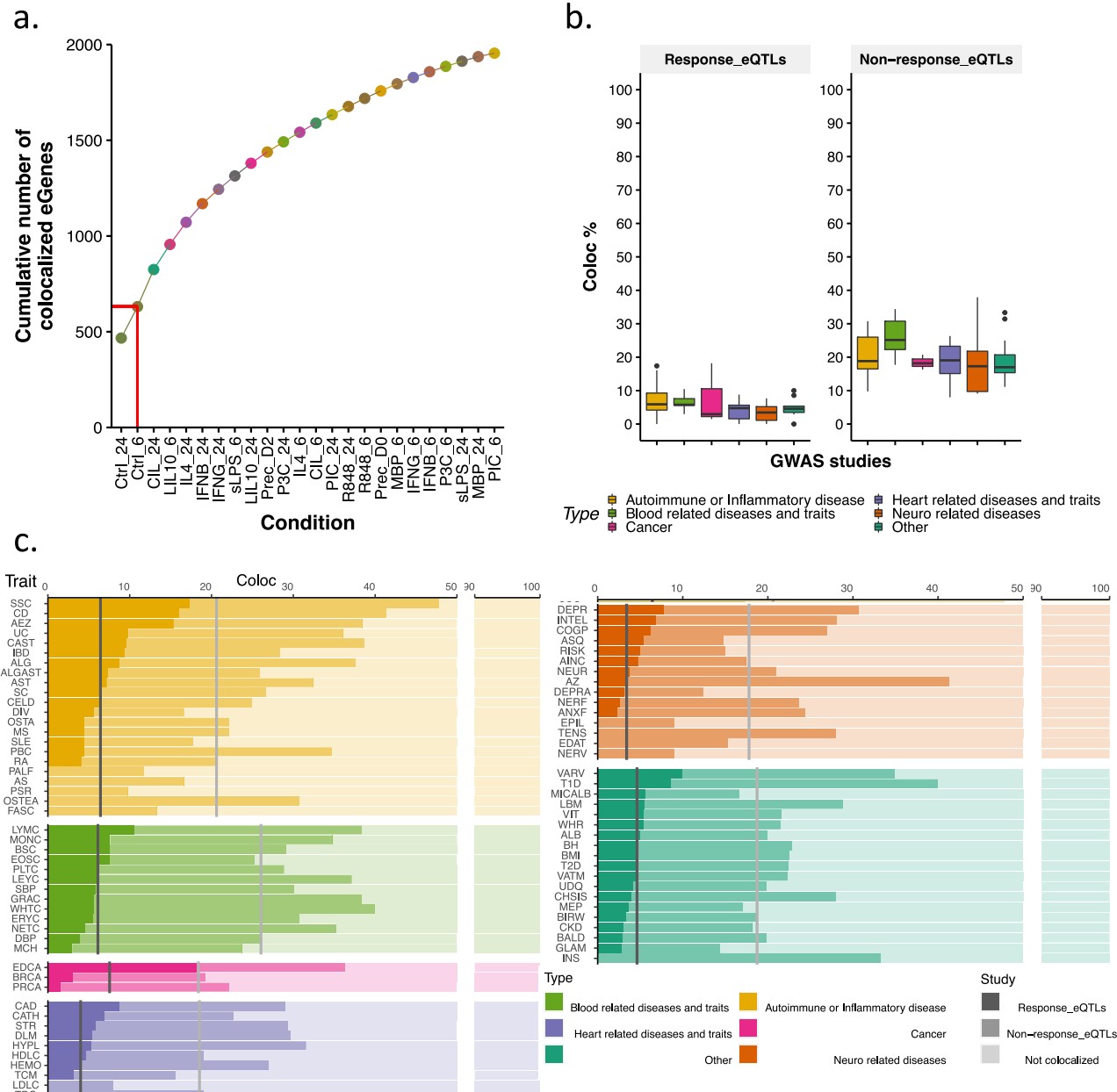

**Fig. 4 | Colocalization of GWAS loci with response and non-response eQTLs across multiple conditions and traits.** **a** Cumulative number of colocalized eGenes seeded in naïve conditions (Ctrl_24, Ctrl_6), with stimulated conditions shown in increasing order. The red line represents the cumulative count of colocalized eGenes exclusively in naive conditions. **b** Boxplots of percentages of GWAS loci colocalized with response eQTLs and non-response eQTLs across 6 main GWAS categories. Data are presented as box plots for each category, faceted by eQTL type (response eQTLs and non-response eQTLs). Box plots represent the median (centre line), the first and third quartiles (bounds of the box), and the whiskers, which extend to the smallest and largest values within 1.5x the interquartile range from the bounds of the box. Data points outside this range are plotted as outliers. The colocalization percentages were calculated as the proportion of colocalized significant regions among all identified significant regions per GWAS study. Sample sizes (n) for each GWAS category are as follows: Autoimmune or Inflammatory disease (n = 22), Blood related diseases and traits (n = 13), Cancer (n = 3), Heart related diseases and traits (n = 11), Neuro related diseases (n = 15), and Other (n = 19). **c** Percentages of GWAS loci colocalized per trait and category. Grey lines indicate the mean percentage of colocalization with response eQTLs across all traits for a specific GWAS category while the light grey line indicates the mean percentage of colocalization with non-response eQTLs.

(*NEU1*) (complex with *CTSA*), an enzyme which cleaves terminal sialic acid residues from substrates such as glycoproteins and glycolipids[38]. The removal of sialic acid from TLR4, a 2,3 sialylated pattern recognition receptor, with the action of *NEU1*[39,40], facilitates LPS recognition and the subsequent downstream activation of NF-κB signalling and inflammatory cytokine production. Outside the cell, *CTSA* has been suggested to play a role in extracellular matrix (ECM) remodelling[41–44]. ECM remodelling is an integral part of several chronic inflammatory

processes including atheromatous plaque formation within vessel walls, and subsequent CAD[45]. The cardiac expression of *CTSA* is upregulated in multiple animal models of myocardial infarction, Type 2 Diabetes and angiotensin II-stimulated hypertrophy[46–49]. Increased expression of *CTSA* has been shown to trigger proteolysis of the extracellular antioxidant enzyme EC-SOD, resulting in higher levels of oxidative stress, myocyte hypertrophy, ECM remodelling, and inflammation[44,50]. We thus hypothesise that rs3827066 increases risk

of CAD and AAA by disrupting a *CTSA* regulatory element, leading to increased expression of *CTSA* and elevated oxidative stress during the ECM remodelling stage of atheromatous plaque formation. Detecting this *CTSA* eQTL only upon stimulation (i.e. reQTL) is consistent with ECM remodelling being induced following a prolonged period of vessel wall inflammation in CAD. Overall, our findings suggest that this CAD-associated risk locus exacerbates oxidative stress in the ECM by abnormally upregulating CTSA following an inflammatory response.

## Discussion

In this study, we used a high-throughput cellular system of human IPSC-derived immune cells to survey inter-individual variation in gene expression across a range of stimulated conditions. We identified 10,170 eGenes resulting from the combination of higher depth expression profiling, and enhanced detection of both condition-specific gene expression and the condition-specific effects of genetic variants. Using a statistical method that accounts for low power, we detected a significant change in genetic effect size between naive and stimulated cells in 23% of all eGenes. For this set of response eQTLs we found that the majority were shared widely across stimulated conditions, with effects found in a single condition constituting a small fraction (1%). We discovered 1955 disease-eQTL colocalizations of which 51% were not detectable in any GTEX tissue suggesting many disease-associated variants may function in a condition-specific manner.

Perhaps surprisingly, our results suggest that a high level of biological replication was at least as important as condition-specific genetic effects in boosting our discovery rate. More generally, it is unclear how many eQTLs could have been detected by a better powered study of naive cells rather than profiling a large number of stimulated conditions. Nonetheless, profiling stimulated cells proved valuable for interpreting disease loci. We found that 21.7% (617) of all disease colocalizations had a different effect size, usually greater, in a stimulated condition compared to naive cells. However, we believe our estimate for the true number of response QTLs is likely to be a lower bound, mainly limited by statistical power. In some cases, the statistical method we used performed aggressive shrinkage of eQTL effect sizes towards zero. For example, we observed colocalization of a *CD80* eQTL after stimulation with sLPS at 6 h that was not confidently colocalized in naive cells. Despite this discrepancy in colocalization confidence between naive and stimulated macrophages, this eQTL was not considered a reQTL by mashr (Supplementary Figs. 9 and 10). We expect that future studies with larger sample sizes will increase the number of eQTL effects that can be confidently considered response eQTLs.

Several recent studies suggest that a smaller than expected fraction of GWAS loci colocalize with an eQTL[3,5] either because genetic effects are often restricted to cell types and conditions that are not causal for the trait[6]. MacroMap represents the most comprehensive characterisation of macrophage eQTLs in different environmental contexts to better understand how genetic variants impact human traits. Our findings highlight the value of reQTLs for expanding our understanding of complex diseases. While their transformative impact on significantly reshaping our comprehension remains limited, reQTLs offer unique insights into the intricate genetic regulation that surpass what naive eQTLs can provide. Their contribution to uncovering previously unexplained disease loci is undeniable, paving the way for further exploration and bringing us closer to a comprehensive understanding of the complex genetic mechanisms underlying the disease.

Despite MacroMap being able to uncover a significant proportion of "missing" disease-relevant eQTLs, we acknowledge several key limitations of our study for post-GWAS prioritisation. eQTL data alone lacks the complete picture of how genetic associations relate to regulatory function. This highlights the need for broader, integrated

methods to fully understand the regulatory roles of disease-associated variants[51]. Furthermore, eQTLs exhibit high context-dependence, varying significantly across different tissues and environmental conditions, thereby limiting the generalisability of our findings. Moreover, the complex nature of epigenetic and transcriptional variation adds layers of intricacy to identifying disease-relevant regulatory elements, something that we partially consider here. Lastly, as with many eQTL studies we assume a single causal variant per locus for colocalization. These assumptions do not accommodate the multifactorial nature of gene regulation, where multiple variants can influence multiple genes and vice versa. This complicates the establishment of true causal relationships between genetic variants and gene expression.

Upcoming large-scale single-cell QTL and response QTL studies of primary tissues and appropriate cell models will further advance our ability to detect disease-relevant genetic effects. Well-powered cell type and context specific QTL studies of molecular traits with different genomic properties (for example, splicing, chromatin accessibility and chromatin interactions) will likely further improve our ability to understand human disease biology.

## Methods

### iPSC lines

No statistical methods were used to predetermine sample size. Human iPSCs lines from healthy donors of European descent were selected from the HipSci project[20] (http://www.hipsci.org) for differentiation to macrophages (see Supplementary Note). All HipSci samples were collected from consenting research volunteers recruited from the NIHR Cambridge BioResource (https://bioresource.nihr.ac.uk/studies/cbr62/; NIHR BioResource Study Code: CBR62; ethical approvals REC 09/H0304/77, V3 15/03/2013 and REC 09/H0304/77, V2 04/01/2013). None of cell lines were reported to be commonly misidentified according to ICLAC v13. Briefly, 315 lines were initially selected and 227 of them (71.6%) were successfully differentiated. RNA-seq libraries were produced for 217 lines and based on quality control 209 unique lines (Supplementary Data 11) (4698 unique RNA-seq libraries across all conditions) were included in the final dataset (Supplementary Fig. 1A, B).

### RNA-seq and quality control

To process the large number of libraries more efficiently, two RNA-seq library construction protocols were utilised, including a modified Smart-seq2 protocol and the NEBnext Ultra II Directional RNA Library kit (further details provided in the supplementary note). However, this resulted in a batch effect due to the different library preparation methods. This effect was included as a covariate in all downstream analyses.

RNA-seq reads (75 bp paired-end) were aligned to the GRCh38 reference human genome and gencode v.27 transcript annotation using STAR_2.5.3a[52]. To quantify gene expression we used feature-Counts v1.5.3[53]. We kept protein-coding and lincRNA genes in all analyses with mean expression >= 0.5 transcripts per million (TPMs) in at least half of the conditions (≥12), resulting in a total of 14,060 genes. To ensure the quality of the samples, we employed several QC metrics. Principal Components Analysis (PCA) was performed per 96-library pool (4 iPSC lines per pool, 24 conditions per iPSC line) to detect sequencing outliers. Non-stimulated or mislabeled labelled stimulated samples were identified and discarded based on pairwise PCA comparisons of each condition with the rest of the conditions, per 96-library pool. Sex incompatibility checks were also performed using the methods described in ref. 54 and 3 iPSC-lines (72 samples) were discarded due to discordant sex annotations. Subsequently, we performed UMAP analysis[55] to cluster the different conditions and wrongly labelled samples that passed PCA filtering were discarded. Finally, we utilised the Match BAM to VCF (MBV) method[56,57] to detect sample swaps and cross contamination between RNA-seq samples. We

discarded 3 iPSC-lines (72 samples) and 63 additional samples due to cross contamination, and corrected the labels for 23 iPSC-lines identified as swaps. We did not observe concordance of genotype-RNA-seq data for 4 lines which we kept in the final dataset for differential expression analysis but discarded from eQTL mapping. Among the 23 swaps, 2 lines were identical with lines already present in the data and were subsequently removed from the dataset.

In total, we discarded 8 iPSC-lines (-3.7% of the successfully differentiated lines) and 510 RNA-seq samples (-9.8%) based on our QC metrics (318 samples based on all QC metrics, 192 from the discarded 8 iPSC-lines, Supplementary Fig. 1C) resulting in a total of 4698 unique RNA-seq libraries across all conditions.

## Variance component analysis
We adopted the same approach that was implemented in ref. 58 to quantify transcriptional variation. In brief, we used a linear mixed model that employed log10(TPM + 0.1) values for the 14,060 genes, with 15 technical confounders (including RunID, Donor, Stimulus Hours, Sex, Library preparation method, Date thawed, Passage number at thawing, Passage EB formation, IPSCs culture time, Total Harvests, Differentiation time No of Days, Purity results, Estimated cell diameter, SD cell diameter, and Differentiation media) fitted as random effects with independent variance parameters $\varphi_\kappa^2$. We measured the variance explained by factor k using the intraclass correlation $\varphi_\kappa^2/(1+\varphi_\kappa^2)$, while the remaining 14 factors were held constant. The standard error of the intraclass correlation was computed using the delta method, with the standard error of the variance parameter estimator.

## Genotypes
GRCh37 imputed genotypes were obtained from the HipSci project[20]. We utilised CrossMap[59] to lift over the variant coordinates from GRCh37 to GRCh38. We then used bcftools to filter the resulting VCF file, retaining only variants with INFO score >0.4 and minor allele frequency (MAF) > 0.05. To address population stratification, we used EIGENSTRAT[60] to calculate genotype principal components (PC) for the retained variants.

## UMAP clustering and visualisation
To visualise the transcriptional variation across conditions, we applied UMAP analysis to the gene expression data. Prior to UMAP, we performed several preprocessing steps on the log-transformed transcripts per million (TPMs). First, we quantile-normalised the log-TPMs to remove technical differences between samples. Next, we applied a rank-based inverse normal transformation to ensure that the gene expression values were normally distributed. Finally, we regressed out (linear regression) the effects of several covariates including runID, donor, library preparation method, sex, purity results, differentiation media, estimated cell diameter, and Differentiation time No Days (Time in days from EB plating until the day of successful harvest) to account for technical variation and batch effects. The resulting UMAP plot provided a low-dimensional visualisation of the transcriptional differences among the different conditions.

## Differential gene expression analysis
DESeq2[22] was used to identify differentially expressed genes between the naive and stimulated conditions, and SVA (surrogate variable analysis)[61] was employed to detect hidden technical variation that could not be captured by our technical covariates. Specifically, we fitted the samples from both the 6 and 24 h time points of the stimulated and naive conditions together and included 10 SVA factors that were determined from the overall sample composition. An interaction term was also included in the model as shown below:

DESeqDataSet(group1_vs_group2,design = ~ X1 + X2 + X3 + X4 + X5 + X6 + X7 + X8 + X9 + X10 + Stimulus + Hours + Stimulus:Hours).

Following this, the same model was fitted without the interaction term and a likelihood ratio test (test = "LRT") was performed to compare the full model (including all SVA factors and the interaction term) to the reduced model (including all SVA factors but not the interaction term).

To identify differentially expressed genes at specific timepoints (either 6 or 24 h) or genes showing different differential expression patterns between time points (interaction term, time point effects), we used the Wald test (test = "Wald", alpha = 0.05) in DESeq2. Finally, we assessed significance at a 5% false discovery rate (FDR) using the Benjamini-Hochberg and kept genes with abs(log2FoldChange) ≥)1.

Gene Ontology (GO) and Reactome pathway enrichment analyses were conducted using the clusterProfiler and ReactomePA R packages. Over-representation of biological processes and pathways among differentially expressed genes (DEGs) was determined using a hypergeometric test. $p$ values were adjusted for multiple testing using the Benjamini-Hochberg method, with terms or pathways considered significant at an adjusted $p$ value (FDR) < 0.05.

## eQTL mapping
We mapped cis-eQTLs within ±1Mbp of the transcription start site (TSS) of each gene using QTLtools v.1.1[56]. Briefly, QTLtools conducts permutations of the expression data for each gene to record the best $p$ value for any SNP in the cis window. The distribution of the best $p$ values follows a beta distribution under the null hypothesis, and QTLtools estimates the parameters of the beta distribution of each gene through maximum likelihood, which depends on the LD structure of the cis region. An adjusted gene-level $p$ value is computed based on the beta distribution for each gene. To correct for multiple testing across all genes, we used the $q$ value R package on the adjusted gene-level $p$ values obtained from 1000 permutations and significance was assessed at 5% FDR ($q$ value < 0.05) to identify genes with at least one significant cis-eQTL ("eGenes"). We included expression PCs (35–50 depended on condition) and 3 genotyping PCs as covariates to correct for technical variation and capture population stratification. To determine the optimal configuration (number of expression PCs per condition) that maximised the number of discoveries (eGenes), we repeated the entire analysis multiple times using different numbers of expression PCs. Multiple independent signals (5% FDR) for a given eGene were identified by forward stepwise regression followed by a backwards selection step implemented in QTLtools (conditional pass).

## Functional enrichment analysis
We performed an enrichment analysis of genomic annotations to investigate the functional implications of our identified eQTLs. Firstly, we utilised Ensembl's Variant Effect Predictor (VEP) and the Ensembl Regulatory Build to annotate the eQTLs. To identify specific genomic annotations enriched among our eQTLs, we used the first stage hierarchical model implemented in PHM[62] (https://github.com/natsuhiko/PHM).

## Response eQTLs using Multivariate Adaptive Shrinkage (mash)
In order to determine which eQTLs in our dataset were truly restricted to stimulation (reQTLs) we used mashr[28] by following the workflow provided by the authors of mashr(https://stephenslab.github.io/mashr/articles/eQTL_outline.html). Initially, we calculated the standard errors of QTL effect sizes (betas) from QTLtools nominal output, which were combined with effect sizes as input data for mash. Our analysis consisted of two subsets of tests: a random subset of 200,000 tests comprising both null and non-null tests, and a more focused "strong" subset that specifically included the lead SNP (lowest $p$-value) per gene across all conditions, emphasising the most impactful associations. While the strong subset of tests was used to learn data-driven

covariance matrices, the random subset of tests was used to estimate mixture weights and scaling coefficients, as well as to learn the correlation structure among null tests. Additionally, we employed the mashr mode "mashr with common baseline," as described here (https://stephenslab.github.io/mashr/articles/intro_mashcommonb aseline.html), by setting Ctrl_24 as our baseline condition and excluding Ctrl_6 from the mash analysis. In the common baseline mode, mashr estimated the deviation of eQTL effect sizes in each alternative condition from that of the baseline condition, taking into account the correlation that arises when comparing all conditions to a common baseline. To execute the analysis, we first fitted the mashr model to the random tests to determine the mixture weights. We then utilised the model fit to compute the posterior mean effect sizes (mash effect sizes) on the best associated SNP per gene for every stimulation condition. We considered significant response eQTLs (reQTLs) gene-SNP pairs with a local false sign rate (lfsr below 0.05. The lfsr is a measure that is stricter than the false discovery rate (FDR) since it not only requires significant discoveries to have a nonzero value, but also to have a consistent sign[29]. To determine the reQTLs that have consistent effects across multiple conditions (shared reQTLs) or function in only a single condition (condition-specific reQTLs), we needed to investigate whether the gene-SNP pair for a particular condition had lfsr <0.05 in the other conditions. This allowed us to quantify the level of sharing of response effects among the different conditions. For instance, if a particular gene-SNP pair was significant in three out of four stimulation conditions, we considered it a shared reQTL across those three conditions. Conversely, if a gene-SNP pair was significant only in one condition, we classified it as a condition-specific reQTL for that particular condition.

### Colocalization

Colocalization analysis was performed with coloc v3.2-1[32] between our eQTL summary statistics and 83 publicly available GWAS summary statistics (either from GWAS catalogue or from UK BioBank GWAS[63]). These GWAS represented 22 immune-mediated, 13 blood-related, 3 cancer, 11 cardiovascular, 15 neurological, and 19 other traits or diseases (Supplementary Data 9). To ensure that we only included datasets with sufficient statistical power, we only considered GWAS datasets that had ten or more genome-wide significant regions ($p \leq 5.0 \times 10^{-8}$). Specifically, to identify significant regions, we created bed files for all SNPs that had ($p \leq 5.0 \times 10^{-8}$) for each GWAS study. We then used the 'bedtools merge -d 500000' command to combine overlapping variants into a single region that spanned all the combined variants. The regions were expanded by 500 kb on either side, and any overlapping regions were merged again. Next, we run coloc on a 2 Mb window centred on each lead eQTL, using default priors and set the colocalization threshold as PP4 > 0.75. The colocalization proportions were calculated as the proportion of colocalized significant regions among all identified significant regions per GWAS.

### Reporting summary

Further information on research design is available in the Nature Portfolio Reporting Summary linked to this article.

## Data availability

Imputed genotype data for the HipSci lines are available from the European Variation Archive (EVA) (https://www.ebi.ac.uk/eva/?eva-study=PRJEB11749) and European Genome–phenome Archive (EGA) (EGAD00010000773). Unprocessed RNA-seq data are available from EGA under study ID EGAS00001002268 (dataset ID EGAD00001015380). Full summary eQTL statistics, raw and processed counts and full colocalization results are available from Zenodo (https://doi.org/10.5281/zenodo.7967759). Details of available results are listed in Supplementary Data 12.

## Code availability

The code used is available at https://github.com/andersonlab/macromap_eqtl.

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

## Acknowledgements

We would like to thank the Sanger Institute Scientific Operations teams and Human Genetics Informatics team for providing sample handling, data generation and computational support to enable the analyses

described in this manuscript. This work was supported by Wellcome Sanger Institute Core funding from the Wellcome Trust (206194, 220540/Z/20/A). The iPSC lines were generated at the Wellcome Sanger Institute, under the Human Induced Pluripotent Stem Cell Initiative funded by a strategic award (WT098503) from the Wellcome Trust and Medical Research Council. N.I.P. was supported by the Early Postdoc Mobility fellowship from the Swiss National Science Foundation (grant number 178005). M.I. supported by core funding from the British Heart Foundation (RG/18/13/33946), NIHR Cambridge Biomedical Research Centre (IS-BRC-1215-20014), BHF Chair Award (CH/12/2/29428) and Cambridge BHF Centre of Research Excellence (RE/18/1/34212).

## Author contributions

N.I.P. performed the analyses and, along with O.E.G., C.A.A. and D.J.G. drafted the manuscript with contributions from all authors. O.E.G. assisted with multiple analyses, independently validated eQTL results and provided feedback to improve the manuscript. J.R. performed the conditional eQTL analysis. N.K. provided statistical feedback on several occasions, assisted with multiple analyses, and conducted analysis on the pilot data. M.I., L.B.V., A.Ts. and C.G. applied the differentiation protocol, performed QC metrics on the differentiated macrophages and carried out the stimulations under the supervision of C.G. A.K. optimised the low-input bulk RNA-seq protocol and prepared the RNA-seq libraries along with M.I. and A.B. A.To. created the web portal to enhance online access to our resources. D.J.G. conceptualised the study while C.A.A. and D.J.G. supervised this work.

## Competing interests

C.A.A. has received consultancy or lectureship fees from Genomics plc, BridgeBio and GlaxoSmithKline. The rest of the authors declare no competing financial or non-financial interests. D.J.G. was an employee of BioMarin and N.I.P. was an employee of GSK at the time the manuscript was submitted.
