## [Transparent Peer Review file · Nature Communications]

Gene expression QTL mapping in stimulated iPSC-derived macrophages provides insights into common complex diseases.

Corresponding Author: Dr Carl Anderson

Version 0:

Reviewer comments:

Reviewer #1

(Remarks to the Author)

The research offers a highly valuable resource enabling eQTL investigations within an extensive cohort of human iPSC-derived macrophages. The authors have shared the substantial dataset (60 GB) on Zenodo. However, the broader community would greatly benefit if the authors could enhance accessibility to the data by making it interactively queryable, with the corresponding URL provided in the manuscript. Additionally, there is room for further clarification of the methods. Some sentences within the Results section might benefit from rephrasing to enhance clarity. The Discussion section was, to a certain extent, repetitive of the Results section. The authors may further discuss the limitation and implications for future studies.

Some specific comments include:

- Although all the eQTL data were provided on Zenodo, this reviewer highly recommends the authors to further enhance accessibility to the data by making it interactively queryable through, for example, Shiny web tools, with the corresponding URL provided in the manuscript. The effort will enhance the impact of the study and benefit the broad community.
- It was not straightforward to find out how to access the “1955 disease-eQTL colocalizations”. A summary of the supplemental spreadsheets and a summary of the datasets on Zenodo will be informative and guide the readers to understand what is available and determine if downloading the full datasets is needed or not.
- The authors cited reference 15 as the source for the iPSC-macrophage differentiation protocol and described that the current protocol involves minor modifications. Although reference 15 did offer more details, it also directed readers to another previously published paper for supplementary information. Furthermore, discerning the precise minor modification from reference 15 was not straightforward. To address this, the reviewer recommends that the authors simply provide all experimental details in the Methods section. Notably, both X-VIVO 15 and StemPro-34 media were described. Given the extensive scope of the study outlined, such a choice is reasonable. Nevertheless, the reviewer suggests the authors to clarify the evaluation process to ascertain whether the differentiation media might introduce confounding variables. Supplementary Figure 1d provided some information, but the implication should be more clearly spelled out.
- Supplementary Figure 5C: The legend needs to be modified. The light purple should indicate “non-timepoint specific reQTLs”. The style of Figure 4 legend seems to have better clarity. Figure 2b and Supplementary Figure 5a may adapt a similar style as to Figure 4.
- The term “condition-specific QTL” was confusing when being mentioned for the first time in line 247. Then, in line 362-363, the authors explained that “To determine the reQTLs that have consistent effects across multiple conditions (shared reQTLs) or function in only a single condition (condition-specific reQTLs).” Line 172 also mentioned “conditionally independent eQTL.” Please consider rephrasing to further enhance clarity.
- The Discussion section is, to some extent, repetitive. For example, the Results section has already discussed that “profiling naive cells at two different timepoints yielded an additional 774 new eQTLs” as mentioned in the Discussion section. Please highlight the key findings, discuss the limitation and the implications for future studies. For example, limitation of eQTL-based post-GWAS prioritization, and limitation of eQTL studies using iPSC-differentiated cells. Have the authors compared their findings with eQTL studies in primary human monocyte-derived macrophages, and what about the similarities and differences?
- Line 58 “profiled their transcriptome via a low-input RNA-seq protocol”: The sentence implies that only D0 and D2 samples

were profiled via the low-input RNA-seq protocol, which seems not the case. The authors may consider rephrasing.

Reviewer #2

(Remarks to the Author)

In this study the authors generated and analyzed a unique eQTL data set spanning 209 genetically unique individuals and 24 immune-related cellular conditions. The data set alone provides an exciting and valuable resource that can be further explored by other scientists in the future. Context-specific eQTL analysis on the newly generated data was appropriately performed using mashr. The authors attempt to demonstrate the relevance of the newly generated context-specific eQTLs by intersecting eQTL data with GWAS data using coloc. However, I am slightly concerned some of the claims made in the paper are unsupported and may require additional analysis or editing. Overall, this manuscript and associated data provides a valuable contribution to the field.

Major concerns

Figure 2a is misleading. Is it correct that multiple testing is performed in each condition, independently? It is unfair to compare the number of genes detected in each GTEx tissue (where multiple testing correction was assessed independently in each tissue) with the total number of genes detected aggregated across 24 eQTL studies (where multiple testing correction was performed independently in each of the 24 studies). This is made worse by claiming the sample size is 209, when the eQTL sample size was effectively 5,208. Relatedly, I believe lines 364-365 of the discussion should be edited.

The authors claim in main text lines 259-261 that “The rise in the number of eGenes with colocalization evidence can be attributed to both the inclusion of more conditions relevant to disease and the increased power due to the substantial level of biological replication in our study.” I agree that both of events are plausible. However, it is also plausible that the rise of eGenes with colocalization evidence is entirely explained by the substantial level of biological replication in our study (ie. due to increased effective eQTL sample size). The authors rely on the following sentence to support their claim: “Disease-colocalizing eGenes showed a significant overrepresentation of reQTL genes”. I believe this claim is very important to one of the main points of the paper and some secondary analyses are warranted. For example: Do response eQTLs originate in genes with more significant associations than non-response eQTLs simply because reQTLs are statistically difficult to detect? If so, are response eQTLs still enriched in disease-colocalizing eGenes after controlling for eGene significance?

Related to the above point, this manuscript emphasizes that this new data allows for discovery of eQTL in diverse contexts that could not be discovered through standard analysis of bulk tissue eqtls such as GTEx. How many additional colocalizations are discovered with this new eQTL data set that are not discovered by colocalization in one of the GTEx tissues?

Minor concerns

In lines 169-171, the authors report: “Across all conditions, 23.4% (2378) of the 170 eGenes had a reQTL in at least one condition with the majority of these (21.9%) have a larger effect following stimulation than in the naïve condition”. This is very surprising and, barring no technical artifacts, would seem to imply genes are more heritable in stimulated conditions than in naïve conditions. Perhaps, this could be a technical artifact of the expression normalization procedure. Do the authors have any explanations why there is such a strong directional effect?

How can it be reconciled that IL4-24 and IL4-6 are viewed as abnormal samples on the UMAP plot with clearly distinct expression profiles (figure 1b), yet highly correlated with prec_D0, prec_D2, ctrl_6, and ctrl_24 on Supplementary Fig. 2a?

In lines 86-86 of the main text the authors claim “identified stimulation as the second most important driver of total expression variation”. I think it should be clarified that “stimulation” here corresponds to “stimulation differentiation time”, as opposed to a categorical variable describing each individual stimulus. Could it be clarified how much expression variation is explained by each individual stimulus?

It is obviously of large importance that the authors publicly release the raw expression data upon release of the publication.

Version 1:

Reviewer comments:

Reviewer #1

(Remarks to the Author)

The authors have addressed all of this reviewer's comments. Congratulations to the authors on this important work.

Reviewer #2

(Remarks to the Author)

I thank the authors for a very thorough response my previous concerns. I have no additional comments.

Point-by-point response to the Reviewer's comments

Reviewer #1

The research offers a highly valuable resource enabling eQTL investigations within an extensive cohort of human iPSC-derived macrophages. The authors have shared the substantial dataset (60 GB) on Zenodo. However, the broader community would greatly benefit if the authors could enhance accessibility to the data by making it interactively queryable, with the corresponding URL provided in the manuscript. Additionally, there is room for further clarification of the methods. Some sentences within the Results section might benefit from rephrasing to enhance clarity. The Discussion section was, to a certain extent, repetitive of the Results section. The authors may further discuss the limitations and implications for future studies.

We express our gratitude to the reviewer for the insightful comments provided. The feedback is greatly appreciated and considered highly valuable in enhancing the quality of our work. We have taken into account each point raised and have addressed them in the subsequent responses. Detailed responses to each comment can be found below.

1. Although all the eQTL data were provided on Zenodo, this reviewer highly recommends the authors to further enhance accessibility to the data by making it interactively queryable through, for example, Shiny web tools, with the corresponding URL provided in the manuscript. The effort will enhance the impact of the study and benefit the broad community.

We appreciate the reviewer's suggestion for enhancing the accessibility of our eQTL data. We have now created a web portal (<https://macromapqtl.org.uk/>) in order to enhance the accessibility to the data by making them interactively queryable. Moreover, our goal is to incorporate our data into eQTL Catalogue (<https://www.ebi.ac.uk/eqtl/>), a project that aims to provide uniformly processed gene expression and splicing QTLs from many available public studies. This would allow for direct comparisons with multiple other eQTL datasets that have been processed through the same pipeline. Furthermore, this will also enable our data to be processed by Open Targets, an innovative, large-scale, public-private partnership that uses human genetics and genomics data for systematic drug target identification and prioritisation. Open Targets integrates publicly available datasets to build and score target-disease associations, thereby enhancing the accessibility and impact of our study.

We hope this approach meets the reviewer's expectations for data accessibility and community benefit. We appreciate the constructive feedback and the opportunity to clarify our approach to data accessibility.

2. It was not straightforward to find out how to access the "1955 disease-eQTL colocalizations". A summary of the supplemental spreadsheets and a summary of the datasets on Zenodo will be informative and guide the readers to understand what is available and determine if downloading the full datasets is needed or not.

We appreciate the reviewer's suggestion for improving the accessibility and understanding of our "1955 disease-eQTL colocalizations". We acknowledge that these data were not included in the initial submission, and we apologise for any confusion this may have caused.

In response to this feedback, we have now included these data as a supplementary table accompanying the manuscript (supplementary table 10) . This table provides a comprehensive overview of the disease-eQTL colocalizations, allowing readers to quickly assess the content and determine if downloading the full datasets is necessary for their purposes. Furthermore, we have added all colocalization analysis data to Zenodo (<https://zenodo.org/records/10797132>) for comprehensive access. This will also allow interested readers to delve deeper into the data, if they wish.

We agree with the reviewer's suggestion to provide summaries of the supplemental spreadsheets and datasets on Zenodo. These tables have been added as supplementary table 12.

3. The authors cited reference 15 as the source for the iPS-macrophage differentiation protocol and described that the current protocol involves minor modifications. Although reference 15 did offer more details, it also directed readers to another previously published paper for supplementary information. Furthermore, discerning the precise minor modification from reference 15 was not straightforward. To address this, the reviewer recommends that the authors simply provide all experimental details in the Methods section. Notably, both X-VIVO 15 and StemPro-34 media were described. Given the extensive scope of the study outlined, such a choice is reasonable. Nevertheless, the reviewer suggests the authors to clarify the evaluation process to ascertain whether the differentiation media might introduce confounding variables. Supplementary Figure 1d provided some information, but the implication should be more clearly spelled out.

We thank the reviewer for the valuable feedback. All the necessary information has now been collected and added to the supplementary note for clarity.

Regarding the use of X-VIVO 15 and StemPro-34 media, we would like to draw the reviewer's attention to Supplementary Figure 1e. This figure demonstrates that there is considerable technical variation due to the media. We understand the importance of clarifying whether the differentiation media introduces confounding variables. The reviewer's suggestion has been taken into account and this has been made clear in the text now. (lines 94 - 100).

"We first explored which technical and biological confounders affected gene expression using variance component analysis (Supplementary figure 1d) and identified that the combined factors of stimulation type and differentiation time, as the second most important driver of the total expression variation (29.3%) after the library preparation method (39.9%). Notably, the library preparation method variation is primarily due to the choice of medium used during myeloid precursor formation (Supplementary Figure 1e)."

We hope that these changes address the reviewer's concerns adequately and welcome any further feedback.

4. Supplementary Figure 5C: The legend needs to be modified. The light purple should indicate "non-timepoint specific reQTLs". The style of Figure 4 legend seems to have better clarity. Figure 2b and Supplementary Figure 5a may adapt a similar style as to Figure 4.

We express our gratitude to the reviewer for their keen observations and suggestions. Corrections have been made to the legend in Supplementary Figure 5C, now indicating "non-timepoint specific reQTLs" in light purple. The style has been adjusted to mirror that of Figure 4 for enhanced clarity. In response to the suggestion of replotting, we have now revised Supplementary Figure 5a.

We appreciate the suggestion to modify the style of Figure 2b. We had carefully chosen each colour in Figure 2b to represent different elements, with the intention of providing a clear and distinct representation for the readers. We are concerned that any alterations might introduce potential confusion. We trust that the reviewer will understand our consideration in maintaining the current style of Figure 2b. We are, as always, open to further suggestions and discussions.

5. The term “condition-specific QTL” was confusing when being mentioned for the first time in line 247. Then, in line 362-363, the authors explained that “To determine the reQTLs that have consistent effects across multiple conditions (shared reQTLs) or function in only a single condition (condition-specific reQTLs).” Line 172 also mentioned “conditionally independent eQTL.” Please consider rephrasing to further enhance clarity.

We would like to thank the reviewer for their insightful comments. We agree that the terminology used can indeed be confusing for readers. In response to this feedback, we enhanced the clarity of the manuscript by defining terms when they first appear in the text. We acknowledge that the term “condition-specific QTL” was used in previous versions of the manuscript and was not corrected throughout the text before submission. We appreciate the reviewer’s keen eye in catching this oversight which has now been corrected. Regarding the term ‘conditionally independent eQTLs’, this is defined previously in the manuscript (please see line 139). *“Secondary and tertiary eSNPs (conditionally independent eQTLs) “*

Once again, we thank the reviewer for their valuable input, which will improve the quality and readability of our manuscript.

6. The Discussion section is, to some extent, repetitive. For example, the Results section has already discussed that “profiling naive cells at two different timepoints yielded an additional 774 new eQTLs” as mentioned in the Discussion section. Please highlight the key findings, discuss the limitations and the implications for future studies. For example, limitation of eQTL-based post-GWAS prioritization, and limitation of eQTL studies using iPS-differentiated cells. Have the authors compared their findings with eQTL studies in primary human monocyte-derived macrophages, and what about the similarities and differences?

We appreciate the reviewer’s insightful comments and suggestions. In response to the reviewer’s observation about repetition, we conducted a thorough review of the Discussion section. Redundant statements, such as the “additional 774 new eQTLs,” have been removed from the Discussion section and retained only in the Results section. Furthermore, we have enriched our discussion by addressing the limitations of our study as suggested by the reviewer. Please see text added below.

“Despite MacroMap being able to uncover a significant proportion of “missing” disease-relevant eQTLs, we acknowledge several key limitations of our study for post-GWAS prioritisation. eQTL data alone lacks the complete picture of how genetic associations relate to regulatory function. This highlights the need for broader, integrated methods to fully understand the regulatory roles of disease-associated variants. Furthermore, eQTLs exhibit high context-dependence, varying significantly across different tissues and environmental conditions, thereby limiting the generalizability of our findings. Moreover, the complex nature of epigenetic and transcriptional variations adds layers of intricacy to identifying disease-relevant regulatory elements, something that we don’t take into account here. Lastly, as in many eQTL studies we assume a single causal variant per locus for colocalization. These assumptions do not accommodate the multifactorial nature of gene regulation, where multiple

variants can influence multiple genes and vice versa. This complicates the establishment of true causal relationships between genetic variants and gene expression. “

Regarding the comment about the limitations using iPS-differentiated cells and their similarities and differences, these topics have been extensively studied in other papers and we think is beyond the scope of our study for several reasons. For instance, the paper by Kilpinen et al. (<https://www.nature.com/articles/nature22403>) outlines the major sources of genetic and phenotypic variation in iPS cells and establishes their suitability as models of complex human traits studies. Another study by Alasoo et al. (<https://www.nature.com/articles/srep12524>) provides transcriptional profiling of macrophages derived from monocytes and iPS cells, identifying a conserved response to LPS and novel alternative transcription. We think that our contribution would be more meaningful by focusing on areas that are less well explored. We hope this provides clarity on why we consider these topics to be outside the scope of our current study. We appreciate the reviewer's understanding and consideration.

7. Line 58 “profiled their transcriptome via a low-input RNA-seq protocol”: The sentence implies that only D0 and D2 samples were profiled via the low-input RNA-seq protocol, which seems not the case. The authors may consider rephrasing.

We have addressed this issue and the entire paragraph describing the generation of the dataset has been revised appropriately.

“During the differentiation process, we collected macrophage precursor cells at day 0 (labelled “Prec_D0”) and day 2 (“Prec_D2”). We used a low-input RNA-seq protocol to profile the transcriptome of these cells, as well as that of unstimulated iPSC-derived macrophages after six and 24 hours. We next perturbed the cells with a panel of ten different stimuli and measured gene expression six and 24 hours after stimulation using the same protocol (Figure 1a)”

We thank the reviewer for bringing it to our attention.

Reviewer #2

In this study the authors generated and analyzed a unique eQTL data set spanning 209 genetically unique individuals and 24 immune-related cellular conditions. The data set alone provides an exciting and valuable resource that can be further explored by other scientists in the future. Context-specific eQTL analysis on the newly generated data was appropriately performed using mashr. The authors attempt to demonstrate the relevance of the newly generated context-specific eQTLs by intersecting eQTL data with GWAS data using coloc. However, I am slightly concerned some of the claims made in the paper are unsupported and may require additional analysis or editing. Overall, this manuscript and associated data provides a valuable contribution to the field.

We appreciate the reviewer's insightful and constructive feedback on our manuscript, and we thank them for finding our work valuable. We have taken into account each point raised and have addressed them in the subsequent responses. Detailed responses to each comment can be found below.

Major concerns

1. Figure 2a is misleading. Is it correct that multiple testing is performed in each condition, independently? It is unfair to compare the number of genes detected in each GTEx tissue (where multiple testing correction was assessed independently in each tissue) with the total number of genes detected aggregated across 24 eQTL studies (where multiple testing correction was performed independently in each of the 24 studies). This is made worse by claiming the sample size is 209, when the eQTL sample size was effectively 5,208. Relatedly, I believe lines 364-365 of the discussion should be edited.

We thank the reviewer and appreciate their thoughtful reflections. We have to mention the challenge of presenting a fair comparison between our study and GTEx. In our manuscript, we thoroughly discuss both the comparison of single conditions in our study with those in GTEx tissues and the aggregated eGene numbers. This approach aims to ensure transparency and clarity for our readers. Our reporting methodology (aggregated number of eGenes) aligns precisely with GTEx's approach as outlined in their 2020 science paper (<https://www.science.org/doi/10.1126/science.aaz1776>), including multiple testing corrections at the tissue level and eGene aggregation across all GTEx tissues. However, we acknowledge the fairness concern raised by the reviewer regarding comparing the aggregated eGene numbers and we edited the lines (367-368) in the discussion as suggested. Moreover, we edited Figure 2a by adding all MacroMap conditions in the plot and we explicitly mentioned the square point in the aggregated number of eGenes.

Regarding comparison of single conditions, we state (lines 144-146) *"Per condition, our eQTL detection rate (22.4% of tested genes, mean over all conditions) was lower compared to GTEx tissues of similar sample size (35% in GTEx tissues of between 180-210 individuals)"*.

When considering the aggregated number of eGenes, we state (lines 158-164) *"Although studying many conditions likely revealed additional eQTLs, the large number of eQTLs we detected is likely also driven by the very high degree of biological replication in our study. We note that even when we compared our control conditions at six and 24 hour timepoints, we detected an additional 774 new eGenes (Supplementary Figure 4c). Thus, generating additional gene expression data from the same individuals increased our power to detect eQTLs relative to other studies of similar sample size."*

Furthermore, lines 378-392 in the Discussion again address the power gained by performing eQTL mapping across so many conditions (biological replicates). We believe this is an honest and fair reflection of why our number of eGenes is higher than one may expect for a study of ~200 individuals.

Respectfully, we disagree with the reviewer regarding the effective sample size. Our study profiles the same individuals and cell types after stimulation at different time points. The effective sample size would be 5208 if we were profiling different individuals and various cell types or tissues. In contrast, GTEx encompasses a broader range of individual compositions, tissues, and actual participant numbers. The power of detection in eQTL studies primarily stems from the number of individuals. To address potential concerns about statistical power, we explicitly state in the manuscript that we consider two metrics: sample size and the number of libraries. We trust that the reviewer will appreciate our thoughtful approach, and we remain open to further suggestions and discussions.

2. The authors claim in main text lines 259-261 that "The rise in the number of eGenes with colocalization evidence can be attributed to both the inclusion of more conditions relevant to disease and the increased power due to the substantial level of biological replication in our study.". I agree that both of events are plausible. However, it is also plausible that the rise of eGenes with colocalization evidence is entirely explained by the substantial level of biological

replication in our study (ie. due to increased effective eQTL sample size). The authors rely on the following sentence to support their claim: “Disease-colocalizing eGenes showed a significant overrepresentation of reQTL genes”. I believe this claim is very important to one of the main points of the paper and some secondary analyses are warranted. For example: Do response eQTLs originate in genes with more significant associations than non-response eQTLs simply because reQTLs are statistically difficult to detect? If so, are response eQTLs still enriched in disease-colocalizing eGenes after controlling for eGene significance?

We thank the reviewer for their comment regarding power issues due to biological replication and stimulation. We agree with them regarding power issues due to biological replication. To this end, we don't assert that the 1323 colocalization events (calculated from 1955 (total) - 632(events found in naive states)) are solely due to stimulation. Instead, we are stringent and report only the 424 events that were identified as reQTLs in the mashr analysis.

To further support claims and to make our claim more robust we conducted additional analyses based on the reviewer's feedback. Indeed, by comparing the p-values between reQTLs and non-reQTLs, it becomes evident that reQTLs originate from eGenes with more significant associations, as illustrated in the figure below.

Next, we hypothesised that if this observation was due to detection power issues, reQTLs/re-eGenes with colocalization evidence would have a high percentage of differential expression (up-regulation), under the assumption that there is an undetected effect which only becomes apparent when gene expression is increasing. We found that 126/424 (~30%) are differentially expressed genes and up-regulated, while the remaining 70% of reQTL genes that colocalize are either differentially expressed but down-regulated (61/424, ~14%) or not differentially expressed at all (237/424, ~56%).

We believe that this supports our claim that colocalizations can be attributed to stimulation since the effects are fixed and these are primarily detected due to genetic variation. We also attempted to follow the reviewer's recommendations regarding controlling for eGene significance. However, we couldn't find any suitable methodology to perform this task. If the reviewer can direct us to a relevant method, we would be pleased to implement it. Nonetheless, we believe that the analysis and results we provide are sufficient to address their comment.

3. Related to the above point, this manuscript emphasizes that this new data allows for discovery of eQTL in diverse contexts that could not be discovered through standard analysis of bulk tissue eqtls such as GTEx. How many additional colocalizations are discovered with this new eQTL data set that are not discovered by colocalization in one of the GTEx tissues?

We appreciate the reviewer's comment, and we would like to direct their attention to lines (309 to 317) in our manuscript. In this section, we explicitly discuss and compare our colocalization findings with GTEx. To address potential power issues across GTEx tissues and enhance biological insights, we perform the comparison and report the results at the study level (MacroMap vs. GTEx level) across all GWAS analyses. Our analysis reveals that 998 eGenes (51% of 1955) were colocalized with higher confidence in our dataset ($PP4 > 0.75$ in MacroMap) compared to the $PP4$ values (< 0.5) across all GTEx tissues (see Supplementary Figure 7). Additionally, 164 eGenes (16.6%) were identified by MashR as reQTLs. We believe that these results adequately address the reviewer's comment.

Minor concerns

4. In lines 169-171, the authors report: “Across all conditions, 23.4% (2378) had a reQTL in at least one condition with the majority of these (21.9%,2228) having a larger effect following stimulation than in the naïve condition”. This is very surprising and, barring no technical artifacts, would seem to imply genes are more heritable in stimulated conditions than in naïve conditions. Perhaps, this could be a technical artifact of the expression normalization procedure. Do the authors have any explanations why there is such a strong directional effect?

We thank the reviewer for bringing this to our attention and we apologise for any inconvenience caused. We should have mentioned in the text that we report and compare the “**absolute**” effect sizes. This means that there is no strong directional effect since effect sizes can be either positive or negative. We have now included the word ‘absolute’ in the text (line 183) to ensure there are no further misunderstandings.

5. How can it be reconciled that IL4-24 and IL4-6 are viewed as abnormal samples on the UMAP plot with clearly distinct expression profiles (figure 1b), yet highly correlated with prec_D0, prec_D2, ctrl_6, and ctrl_24 on Supplementary Fig. 2a?

We thank the reviewer for their insightful observation regarding the apparent discrepancy between the UMAP plot and Supplementary Fig. 2a in our study and the opportunity to provide a more comprehensive explanation.

The first dimension, **UMAP_1**, captures the majority of expression variation across all samples, aligning with the levels seen in the heatmap in Supplementary Fig. 2a. Notably, IL4 samples, controls, and precursors share similar UMAP_1 coordinates as reflected in the heatmap. However, **UMAP_2** segregates samples based on specific gene expression patterns that differentiate IL4 from the controls. It highlights a small subset of differentially expressed genes responsible for this separation. These genes play a pivotal role in distinguishing IL4 samples from controls. Although IL4-24 and IL4-6 samples exhibit subtle yet functionally relevant changes in gene expression, these alterations might not significantly impact the overall average expression variation patterns (as observed in the heatmap). However, they do alter the samples’ positions in the UMAP space. We believe that the UMAP plot provides finer resolution by capturing specific gene expression differences, while the heatmap reflects global expression variation patterns. We hope this explanation satisfies the reviewer’s inquiry.

6. In lines 86-86 of the main text the authors claim “identified stimulation as the second most important driver of total expression variation”. I think it should be clarified that “stimulation” here corresponds to “stimulation differentiation time”, as opposed to a categorical variable describing each individual stimulus. Could it be clarified how much expression variation is explained by each individual stimulus?

We thank the reviewer for bringing this to our attention. In the text, we have now changed it to “the combined factors of stimulation type and differentiation time, as the second most important driver of the total expression variation” as this is correct (please see lines 94-96). In the figure below, we provide an analysis of how much expression variation is explained per stimulation differentiation time. Additionally, we considered a reasonable approach to measure expression variation by defining the conditions with the highest number of differentially expressed genes as those that explain most of the variance. We conducted a simple

correlation analysis between the number of differentially expressed genes and the variance explained, and we found that the correlation between them is 0.84, as

expected.

7. It is obviously of large importance that the authors publicly release the raw expression data upon release of the publication.

We appreciate this comment regarding data accessibility. We have now included the raw expression data on Zenodo (<https://zenodo.org/records/11563707>), and we have also created a portal, as suggested by Reviewer #1, to enhance data accessibility (<https://www.macromapqtl.org.uk/>). Moreover, upon publication, bam files will be released in EGA.